# Anti-Neuroinflammatory Effect of Alantolactone through the Suppression of the NF-κB and MAPK Signaling Pathways

**DOI:** 10.3390/cells8070739

**Published:** 2019-07-18

**Authors:** Liwei Tan, Jinsheng Li, Yeye Wang, Rui Tan

**Affiliations:** 1School of Materials Science and Engineering, Southwest Jiaotong University, Chengdu 610031, China; 2College of Life Science and Engineering, Southwest Jiaotong University, Chengdu 610031, China; 3College of Medicine, Southwest Jiaotong University, Chengdu 610031, China

**Keywords:** alantolactone, neuroinflammation, central nervous system damage, cerebral ischemia-reperfusion injury, NF-κB and MAPK signaling pathways

## Abstract

Neuroinflammation is a major cause of central nervous system (CNS) damage and can result in long-term disability and mortality. Therefore, the development of effective anti-neuroinflammatory agents for neuroprotection is vital. To our surprise, the naturally occurring molecule alantolactone (Ala) was reported to significantly inhibit tumor growth and metastasis as a result of its excellent anti-inflammatory effects. Thus, we proposed that it could also act as an anti-neuroinflammatory agent. Thus, in this study, a coculture system of BV2 cells and PC12 cells were used as an in vitro neuroinflammatory model to investigate the anti-neuroinflammatory mechanism of Ala. The results indicated that Ala downregulated the expression of proinflammatory factors by suppressing the nuclear factor kappa light-chain enhancer of activated B cells (NF-κB) and mitogen-activated protein kinase (MAPK) signaling pathways. Further evaluation using a middle cerebral artery occlusion and reperfusion (MCAO/R) rat model supported the conclusion that Ala could (1) alleviate cerebral ischemia-reperfusion injury; (2) reduce neurological deficits, cerebral infarct volume, and brain edema; and (3) attenuate the apoptosis and necrosis of neurons. In sum, Ala demonstrates anti-neuroinflammatory properties that contribute to the amelioration of CNS damage, and it could be a promising candidate for future applications in CNS injury treatment.

## 1. Introduction

Central nervous system (CNS) damage, including that caused by stroke, traumatic brain injury, and neurodegenerative disease, is the most prevalent cause of long-term disability and death [1,2]. A growing body of research has suggested that neuroinflammation is a major mechanism of CNS damage [3]. Unfortunately, currently available agents that target neuroinflammation have failed to achieve significant clinical results in the amelioration of CNS injury [4]. Therefore, it is urgent to discover effective anti-neuroinflammatory agents that have neuroprotective effects, especially from natural products [5,6].

The neuroinflammation mechanism is initiated by a large number of proinflammatory factors that are produced by immune effector cells derived from brain tissue [7,8]. The activation of Toll-like receptors in immune effector cells causes the upregulated expression and release of proinflammatory factors in neighboring cells, leading to the cascading amplification of inflammation, which ultimately results in neuroinflammation [9]. Therefore, decreasing the production and release of proinflammatory factors is an effective way to inhibit neuroinflammation. From some available research reports, it could be found that two signaling pathways stand out as being strongly associated with the production of proinflammatory factors: Nuclear factor kappa light-chain enhancer of activated B cells (NF-κB) [10] and mitogen-activated protein kinase (MAPK) [11].

Because it is responsible for the transcription of several proinflammatory cytokines, chemokines, and adhesion molecules, the NF-κB signaling pathway plays a crucial role in many diseases that involve dysregulation of the inflammatory response [12,13]. The NF-κB signaling pathway consists of a series of intracellular secondary reactions [14,15]. First, extracellular stimuli give rise to the phosphorylation and proteasomal degradation of the nuclear factor of the kappa light polypeptide gene enhancer in B-cells inhibitor, alpha (IκBα) [16,17]. Then, the p65 subunit from the phosphorylated and dissociated IκBα translocates to the nucleus. The p65/p50 subunit is the most common heterodimer of the NF-κB complex, and p65 is the subunit of the NF-κB complex, which can regulate the binding of p50 with DNA to activate the downstream reaction. In this process, p65 is critical because of its multifunctional domain, which contains not only a transcriptional activation region that participates in the initiation of gene transcription but also a special transactivating structural domain (TAD) that regulates downstream target genes [18]. Furthermore, the activated p65 is no longer inhibited from translocating to the nucleus, where it leads to the expression of downstream proinflammatory genes, including cyclooxygenase (COX)-2, inducible nitric oxide synthase (iNOS), interleukin (IL)-1β, IL-6, and so on [19,20]. Second, the MAPK signaling pathway has been reported to be involved in cell proliferation, differentiation, and apoptosis [21]. The predominating effect of the MAPK signaling pathway is the activation of three distinct cascades, namely the extracellular signal-related kinases (ERKs), c-Jun N-terminal kinases (JNKs), and p38 pathways [21]. Subsequently, the transcription factor activator protein (AP)-1 is activated and affects the expression of relevant genes [22]. Finally, the expression of proinflammatory factors promotes the generation of neuroinflammation. Therefore, the regulation of protein activity in the NF-κB and MAPK signaling pathways may be the key to reduce anti-neuroinflammation.

Similarly to CNS diseases, inflammation promotes the growth and metastasis of tumors. Surprisingly, alantolactone (Ala), a sesquiterpene lactone isolated from *Inula helenium*, has been proven to be able to have an inhibitory effect on the growth of tumors by causing the significantly downregulated expression of proinflammatory factors [23,24,25]. Moreover, Ala induces apoptosis and suppresses the migration of MCF-7 human breast cancer cells by targeting the MAPK and NF-κB signaling pathways [26]. Thus, it is reasonable to propose that Ala might be a potential anti-neuroinflammatory agent for CNS injury therapy. Moreover, recent research has suggested that Ala possesses high blood–brain barrier (BBB) permeance efficiency, which makes Ala a feasible candidate for CNS treatment [24,27].

In this study, we explored the anti-neuroinflammatory properties of a natural small-molecular agent (Ala) in vitro and in vivo and further to make a detailed investigation of its pharmacodynamic effects and mechanisms. As shown in Scheme 1, lipopolysaccharide (LPS) was used to initiate the inflammatory response and induce the in vitro microglia cell (BV2 cell) neuroinflammation model. The relative expression of proinflammatory factors was detected by real-time quantitative PCR (QPCR), western blot, and enzyme-linked immunosorbent Assay (ELISA). Since neuroinflammation is the most important mechanism involved in cerebral ischemia-reperfusion injury, an in vivo evaluation of Ala was performed using a middle cerebral artery occlusion and reperfusion (MCAO/R) rat model. Overall, the results from this study demonstrated that the compound Ala exerted anti-neuroinflammation effects that could ameliorate CNS injury by affecting the MAPK and NF-κB signaling pathways.

## 2. Materials and Methods

### 2.1. Materials

Alantolactone (Ala, purity >98%) was obtained from Pusi Biotechnology Co., Ltd. (Cheng Du, China). High-glucose Dulbecco’s Modified Eagle Medium (DMEM), fetal bovine serum (FBS), penicillin, and streptomycin were provided by Gibco Life Technologies (Grand Island, NY, USA). LPS (*Escherichia coli* serotype O111:B4), 3-(4,5-dimethylthiazol-2-yl)-2,5-diphenyltetrazolium bromide (MTT), 4′,6-diamidino-2-phenylindole (DAPI), and 2,3,5-triphenyltetrazolium chloride (TTC) were purchased from Sigma-Aldrich (Saint Louis, MO, USA). For western blot assays, antibodies against IκBα, phosphorylated-IκBα (p-IκBα), p-p38, p-ERK, p-JNK, COX-2, iNOS, and glyceraldehyde-3-phosphate dehydrogenase (GAPDH) and horseradish peroxidase (HRP)-conjugated secondary antibodies were obtained from Cell Signaling Technology (Danvers, MA, USA). Anti-p65 antibody, anti-AP-1 antibody, Cy3-conjugated secondary antibody, and fluorescein isothiocyanate (FITC)-conjugated secondary antibody were purchased from Boster (Wuhan, China). For QPCR detection, A TRIzol extraction kit was obtained from Sigma-Aldrich (Saint Louis, MO, USA), and a PrimeScriptTM RT reagent kit with gDNA Eraser was purchased from TaKaRa (Tokyo, Japan). SYBR Green PCR Master Mix was purchased from Thermo Fisher Scientific (Waltham, MA, USA). A KeyFluor488-EdU kit and an Annexin V-FITC/PI Apoptosis Detection Kit were obtained from Keygen Biotech (Jiangsu, China). BD Biosciences (San Jose, CA, USA) provided the Cell Cycle and Apoptosis Analysis Kit. ELISA kits for IL-1β, IL-6, tumor necrosis factor (TNF)-α, and prostaglandin E2 (PGE2) were purchased from Elabscience (Wuhan, China). Griess reagent for nitric oxide (NO) was purchased from Sigma-Aldrich (Saint Louis, MO, USA).

The BV2 and PC12 cell lines were supplied by the Cell Bank of the Chinese Academy of Sciences (Shanghai, China) and the American Type Culture Collection (ATCC; Manassas, VA, USA), respectively.

Male Sprague Dawley (SD) rats (280–300 g) were supplied by Dashuo Biotechnology Co., Ltd. (Chengdu, China). The rats were housed at a temperature of 20 ± 2 °C with a relative humidity of 50–60% and 12-h light/dark cycles. They acclimatized for 2 weeks prior to the experiment. The protocol was authorized by the Institutional Animal Care and Use Committee of Chengdu Military General Hospital.

### 2.2. Cell Culture and Cell Coculture

BV2 and PC12 cells were cultured in high-glucose DMEM with 10% heat-inactivated FBS and 10% FBS, respectively, penicillin (100 U/mL), and streptomycin (100 μg/mL). BV2 and PC12 cells were set in an incubator at 37 °C with a humidified atmosphere of 5% CO_2_. In the coculture system, PC12 cells (2 × 10^5^/well) were incubated on the bottom of the wells in a 6-well plate, and BV2 cells (1 × 10^5^/well) were incubated and then grown in culture inserts (pore size = 0.4 μm; Corning, NY, USA).

### 2.3. RNA Extraction and QPCR

For QPCR analysis, the BV2 cells were pretreated with the indicated concentrations of Ala for 30 min before the addition of LPS (100 ng/mL). Total mRNA was extracted from cells through TRIzol extraction. Both the amount and purity of the RNA preparation were confirmed by measuring the absorbance ratio at 260/280 nm. Total RNA (1 μg) was converted to cDNA using a PrimeScript^TM^ RT reagent kit with gDNA Eraser and PCR amplification followed by an ABI Step One Plus instrument and software (Applied Biosystems, Foster City, CA) using SYBR Green PCR Master Mix. The RNA levels of the target genes were normalized by β-actin according to the 2^−△△Ct^ method. Each procedure was performed in triplicate independently to ensure minimal bias. The primers used in this study were as follows:

TNF-α F:5′-CAGGCGGTGCCTATGTCTC-3′ and R: 5′-CGATCACCCCGAAGTTCAGTAG-3′;IL-1β F: 5′-GCAACTGTTCCTGAACTCAACT-3′ and R: 5′-ATCTTTTGGGGTCCGTCAACT-3′;IL-6 F: 5′-TAGTCCTTCCTACCCCAATTTCC-3′ and R: 5′-TTGGTCCTTAGCCACTCCTTC-3′;iNOS F: 5′-GTTCTCAGCCCAACAATACAAGA-3′ and R: 5′-GTGGACGGGTCGATGTCAC-3′;COX-2 F: 5′-TGAGCAACTATTCCAAACCAGC-3′ and R: 5′-GCACGTAGTCTTCGATCACTATC-3′;β-actin F: 5′-GGCTGTATTCCCCTCCATCG-3′ and R: 5′-CCAGTTGGTAACAATGCCATGT-3′

### 2.4. Cell Viability

Cell viability was estimated by the MTT assay. BV2 cells or PC12 cells were cultured in 96-well plates (5 × 10^3^/well) at 37 °C. After 24 h, one of five Ala concentrations (0.5, 1, 2, 3, 5 μM) was added to each well, and the same volume of dimethyl sulfoxide (DMSO) was added to a well as a negative control. Then, the cells were incubated for 20 h. Then, 5 mg/mL MTT (10 µL) was added to each well, and they were incubated for another 4 h at 37 °C. The media were carefully removed. The formazan crystals were dissolved in 100 μL DMSO, and absorbance was determined at 490 nm with an ELISA reader (MultiskanEX, Lab systems, Helsinki, Finland). All sample experiments were performed in triplicate and repeated three times, independently.

### 2.5. Cell Proliferation Assay

Cell proliferation was assessed by the keyFluor488-EdU kit according to the manufacturer’s protocol. Briefly, BV2 cells were treated with Ala (2 μM) for 30 min and then stimulated with LPS (100 ng/mL) for another 24 h. Then, EdU (10 μM) was added to the plate for 2 h. After they were fixed with 4% paraformaldehyde, the cells were incubated with a Click-iT reaction cocktail for 30 min and then another 10 min for nuclear staining by DAPI. Cell proliferation was identified by green staining. For quantification analysis, five fields per sample were chosen randomly for statistical analysis. All sample experiments were repeated three times, independently [28].

### 2.6. Flow Cytometry

The BV2 cells were treated with Ala (2 μM) for 30 min and then stimulated with LPS (100 ng/mL) for another 24 h. The BV2 cell cycle was detected using Cell Cycle and Apoptosis Analysis Kit staining with 50 μg/mL propidium iodide (PI, 400 μL) according to the standard operating manual. Here, 1 × 10^5^ cells per sample were analyzed. Intracellular DNA content was analyzed using a FACSCalibur flow cytometer (BD Biosciences, San Jose, CA, USA). In the coculture system, Ala (2 μM) was added to the culture insert and left for 30 min before the addition of LPS (100 ng/mL), after which the culture was left for another 24 h. Only the PC12 cells were incubated on the bottom of the wells, while BV2 cells were grown in culture inserts. LPS (100 ng/mL) was added to the culture insert. The purpose of this setup was to prevent the effects of LPS on PC12 cells. After being processed, the PC12 cells were collected, rinsed, and resuspended in PBS. Next, the resuspended PC12 cells were stained with annexin V and PI using the Annexin V-FITC/PI Apoptosis Detection Kit as described in the operation manual to determine the ratio of PC12 cells that experienced apoptosis and necrosis. Briefly, PC12 cells in PBS were centrifuged and resuspended with 500 μL of binding buffer, and 5 μL of annexin V–FITC and 5 μL of propidium iodide (PI) were added to the suspension. The early apoptotic cells (the annexin V-positive and PI-negative cells), the late apoptotic cells (the annexin V-positive and PI-positive cells), the necrotic cells (the annexin-V negative and PI-positive cells), and normal cells (the annexin-V negative and PI-negative cells) were counted separately using a FACSCalibur flow cytometer [29]. Here, 2 × 10^4^ cells per sample were analyzed. Annexin V–FITC and PI could directly stain the PC12 cells on the bottom of the wells after mixing with the binding buffer. Images were obtained by a confocal microscope (EVOS^®^ FLoid^®^ cell-imaging station, Thermo Fisher Scientific, Waltham, MA, USA). The BV2 cell cycle was detected by staining using the Cell Cycle and Apoptosis Analysis Kit with 50 μg/mL of PI (400 μL) according to the standard operating manual. Intracellular DNA content was analyzed using a FACSCalibur flow cytometer (BD Biosciences, San Jose, CA, USA). All sample experiments were performed in triplicate and repeated three times, independently.

### 2.7. Immunofluorescent Staining

Briefly, BV2 cells treated with Ala (2 μM) for 30 min were stimulated with LPS (100 ng/mL) for another 1 h. After that, the cells were fixed with 4% paraformaldehyde, permeabilized with 0.1% Triton X-100 for 10 min, and blocked with 5% bovine serum albumin (BSA) for 30 min. Immunofluorescence detection of rabbit anti-NF-κB p65 antibody and anti-AP-1 antibody was carried with Cy3-conjugated secondary antibody and FITC-conjugated secondary antibody. For nuclear staining, the cells were incubated with DAPI. The fluorescence intensities were quantified using Image J software (National Institutes of Health, Bethesda, MD, USA). The round coverslips were finally washed and mounted for microscopic examination. Five fields per sample were chosen randomly for statistical analysis. All sample experiments were repeated three times, independently.

### 2.8. Western Blot

Total protein content was isolated from BV2 cells lysed with radio immunoprecipitation assay (RIPA) buffer (Invitrogen, Carlsbad, CA). Equal amounts of proteins were separated by Sodium dodecyl sulfate (SDS)-polyacrylamide gel electrophoresis. After that, the proteins were transferred to polyvinylidene fluoride (PVDF) membranes, which were blocked with 5% nonfat milk for 1 h at room temperature and incubated with primary antibodies, including those against COX-2, iNOS, IκBα, p-IκBα, p-p65, p-ERK, p-p38, p-JNK, and GAPDH, followed by incubation with HRP-conjugated secondary antibody for 1 h. The blots were visualized through densitometry via Image J software and the ECL-Plus detection system (Santa Cruz Biotechnology, Santa Cruz, CA). All sample experiments were repeated three times, independently.

### 2.9. Proinflammatory Cytokines, NO, and PGE_2_ Evaluation

Briefly, BV-2 cells were plated in 96-well plates (5 × 10^3^ cells/well) and induced by Ala (2 μM) for 30 min. Then, LPS (100 ng/mL) was added to the cells (except for the negative control cells): Cells with and without LPS were incubated for 24 h. The relevant proinflammatory factor levels in the cultured supernatants were evaluated by the ELISA kits according to the manufacturer’s instructions, and the molecules screened included IL-1β, IL-6, TNF-α, and PGE_2_. Accumulated nitrite was measured in the cell supernatants using a Griess reagent to detect the concentration of NO. The plates were read using a BioTek plate reader immediately (MultiskanEX, Lab systems, Helsinki, Finland). All sample experiments were performed in triplicate and repeated three times, independently.

### 2.10. Animal Model of Cerebral Ischemia-Reperfusion Injury

Rats weighing 280–300 g were purchased from Chengdu Dashuo and were used in this experiment according to Wang’s method [30]. The rats were kept in a thermostatic chamber with freely available food and water. Thread embolization was carried out to establish the rat MCAO model. Briefly, rats were first injected with choral hydrate (10%, 100 g/mL) through intraperitoneal injection. After that, rats were placed in the supine position, and the right common carotid artery was exposed by median jugular incision. The prepared nylon suture was inserted into the external carotid artery and introduced to the middle cerebral artery. The nylon suture was removed after occlusion for 2 h. Subsequently, rats were separated into four groups (24 per group): Sham treatment (with an equal volume of propylene glycol), sham treatment with Ala (20 mg/kg) (Ala group), MCAO with reperfusion (MCAO/R group), and Ala intraperitoneal administration (20 mg/kg) immediately after MCAO and before reperfusion (MCAO/R + Ala group). Twenty-four hours after MCAO, neurological function and the area of cerebral infarction were assessed. In addition, total rat brain tissue was taken from each group for ELISA analysis. Hematoxylin and eosin (HE) staining, a terminal deoxynucleotidyl transferase-mediated 2′-Deoxyuridine 5′-Triphosphate (dUTP) nick-end-labeling (TUNEL) assay, and Nissl staining were performed to identify pathological changes in other brains.

### 2.11. Effect on MCAO/R Rat Model

About 24 h after Ala administration, neurological function was evaluated as follows. According to Zea Longa’s method [31], neurological test scores were categorized into 5 grades: 0, no neurological deficit; 1, unable to fully extend right forepaw upon lifting the whole body by the tail; 2, circling to the right; 3, falling to the right; and 4, unable to walk spontaneously and reduced levels of consciousness.

Infarct volume was measured on the basis of TTC staining. Infarct regions were indicated by pale unstained sections, and normal tissue was indicated by red-stained sections. After that, the slices were photographed, and the infarct regions of the hemispheres were analyzed by a morphological image analysis system (Jie Da software, Jiangsu, China). Infarct volume was calculated according to the following formula:(1)$$V=\sum_{i=1}^{n-1}\frac{\left(A_{i}+A_{i+1}\right)}{2}\times h$$
where *V* is the volume of the fraction, *A_i_* is the infarct area of each slice, and *h* is the slice thickness.

Brain tissue was obtained from MCAO/R and normal rats after the rats were sacrificed. The wet–dry method was applied to calculate brain water content (BWC). After weighing the fresh samples, brains were dried overnight at 105 °C and weighed again (dry weight), and the total brain water content was calculated by
(2)$$BWC\%=\frac{W-D}{W}\times 100\%$$
where *BWC* is brain water content, *W* is wet weight, and *D* is dry weight. All sample experiments were repeated six times, independently.

### 2.12. Histopathological Study

About 24 h after Ala administration, brains were collected and fixed with 4% paraformaldehyde. Morphological features in the hippocampi and penumbrae were observed by HE-staining the samples, as described above. After they were fixed, dehydrated, and embedded in paraffin, the sample sections (5 μm) were stained by HE and examined by microscope (Olympus, Tokyo, Japan) to observe neuron damage. Five fields were examined.

Morphologic changes in the MCAO/R rat model were evaluated by Nissl staining. After a standard tissue preparation procedure (fixing, dehydrating, embedding in paraffin), tissue sections (5 μm slices) were washed in cold water and stained with 1% toluidine blue. An optical microscope was used to observe each slice.

TUNEL staining was applied using an in situ apoptosis detection kit (KeyGen, Jiangsu, China) according to the standard protocol. Before TUNEL staining, the tissues were processed as above. TUNEL-stained sections were examined by a fluorescence microscope (Olympus IX71; Olympus, Tokyo, Japan). The total number of cells and the number of TUNEL-positive cells were counted in each field. The percentage of TUNEL-positive cells was calculated as follows: TUNEL-positive cells (%) = number of positive cells/number of total cells × 100%. Three sections per rat were used in this experiment. All sample experiments were repeated six times, independently.

### 2.13. Proinflammatory Cytokine Evaluation of the MCAO/R Rat Model

Twenty-four hours after Ala administration, total rat brain tissues were taken from each group for ELISA analysis. The ELISA kits were the same as those used for cell detection and included IL-1β, IL-6, TNF-α, and PGE_2_ ELISA kits and Griess reagent. The plates were immediately read using a BioTek plate reader (MultiskanEX, Lab systems, Helsinki, Finland). All sample experiments were performed in triplicate and repeated six times, independently.

### 2.14. Statistical Methods

All experimental data are expressed as the mean ± SD. Single comparisons were tested by one-way analysis of variance (ANOVA), and *p* < 0.05 indicates a statistically significant difference. Prism version 6.0 (GraphPad software, La Jolla, CA) was used to perform statistical analyses.

## 3. Results

### 3.1. Ala Suppressed the Proliferation of LPS-Activated BV2 Cells

To examine the effect of Ala administration on the proliferation of LPS-stimulated BV2 cells, fluorescence microscopy and flow cytometry analysis were performed. As shown in Figure 1A, cell viability was not affected after 24 h of 2 μM of Ala treatment. Simultaneously, we observed that 100 ng/mL of LPS induced the development of ameboid BV2 cells, but the morphological change was partially suppressed by Ala pretreatment (Figure 1B). The result of the keyFluor488-EdU kit assay suggested that LPS activated the BV2 cells and improved cell proliferation, but this was weakened by Ala treatment of the culture. When BV2 cells were treated by Ala alone for 24 h, cell proliferation was also inhibited (Figure 1C). In addition, we detected the stages of the cell cycle by flow cytometry for the four cell groups (i.e., treated with DMSO, Ala, LPS, or LPS and Ala). Figure 1D exhibits that, compared to the DMSO treatment, the 100-ng/mL LPS treatment significantly decreased the percentage of BV2 cells in the G1 phase and increased the proportion of BV2 cells in the S phase. In contrast, after treatment with Ala, a considerable proportion of the cells were in the G1 phase, and the proportion of cells in the S phase decreased significantly. As a whole, the results in Figure 1 illustrate that Ala attenuated LPS-activated cell proliferation and inhibited the G1 phase of the cell cycle in BV2 cells.

### 3.2. Ala Attenuated Proinflammatory Cytokine Release from LPS-Activated BV2 Cells

Figure 2 clearly shows the presence of proinflammatory cytokines. In BV2 cells that were pretreated with Ala (0.5, 1, or 2 μM) for 30 min and stimulated by LPS for another 6 h, the expression of IL-1β and IL-6 decreased in a dose-dependent manner. However, the level of TNF-α decreased only slightly (Figure 2A). QPCR was performed by extracting total RNA to identify the differential expression of proinflammatory factors after cells were treated with 2 μM of Ala for 6 and 24 h. After 6 h, compared to LPS-activated BV2 cells without Ala (Figure 2B), the expression of IL-1β and IL-6 was dramatically decreased in LPS-activated and Ala-treated BV2 cells. TNF-α expression decreased slightly within 6 h (Figure 2B). Furthermore, Ala still played an inhibitory role in LPS-activated BV2 cells after 24 h: However, the inhibitory effect of Ala on TNF-α expression was weak (Figure 2B). We collected the cultured supernatants from BV2 cells 24 h after LPS administration for ELISA analysis to examine the levels of proinflammatory cytokines, including IL-1β, IL-6, and TNF-α. The levels of these proinflammatory cytokines were higher in the LPS group than in the LPS + Ala group (Figure 2C). These results demonstrate that LPS promoted the increased expression of proinflammatory cytokines in BV2 cells and that Ala treatment attenuated this increase.

### 3.3. Effect on COX-2/PGE_2_ and iNOS/NO Expression

To further evaluate the effect of Ala intervention on the expression of proinflammatory factors, we assessed the relative gene and protein expression by QPCR and western blot. Compared to the DMSO group, the expression of COX-2 and iNOS increased in the LPS group and the LPS + Ala group 24 h after the addition of LPS. The expression levels of both COX-2 and iNOS genes were higher in the LPS group than in the LPS + Ala group (Figure 3A). We also examined the expression of COX-2 and iNOS proteins 24 h after the addition of LPS. As demonstrated in Figure 3B, the expression of COX-2 and iNOS proteins were potentiated in the LPS group but inhibited by Ala treatment. PGE_2_ and NO, as downstream proinflammatory factors of COX-2 and iNOS, were also assessed by ELISA analysis. The levels of PGE_2_ and NO had trends that were similar to the expression of COX-2 and iNOS (Figure 3C). Our findings clearly show that Ala suppressed the expression of COX-2/PGE_2_ and iNOS/NO in LPS-activated BV2 cells.

### 3.4. Effect on NF-κB and MAPK Signaling Pathways

Activation of NF-κB leads to its translocation to the nucleus, where it mediates the transcriptional regulation of proinflammatory genes. Two proteins linked with NF-κB were evaluated: IκBα and p65. Three MAPK-related molecules—p38, JNK, and ERK—were significantly phosphorylated, as detected by western blot. These proteins play crucial roles in the MAPK signaling pathway by sending inflammatory signals. As shown in Figure 4A, 1 h after stimulation by LPS, BV2 cells contained p-IκBα and p-p65, and IκBα was degraded. Treatment of BV2 cells with Ala for 30 min followed by stimulation with LPS for another 1 h suppressed the degradation of only IκBα. LPS rapidly activated the phosphorylation of JNK and ERK from the MAPK signaling pathway, while 2 µM of Ala markedly suppressed these LPS-induced changes. However, LPS and Ala had no effect on p38 phosphorylation (Figure 4B). The intensities of these protein bands were further quantified using Image J software (Figure 4C). To better understand the mechanisms of the NF-κB and MAPK signaling pathways, we investigated the location of p65 and AP-1: In the nucleus, these proteins participated in the regulation of inflammatory factor expression. The result demonstrated that both p65 and AP-1 translocated to the nucleus in LPS-activated BV2 cells (Figure 4E,F). In BV2 cells treated with Ala after LPS, most of the p65 and AP-1 proteins were located in the cytoplasm (Figure 4E,F). These findings provide evidence that Ala inhibits the nuclear translocation of NF-κB and AP-1 in LPS-activated BV2 cells.

### 3.5. Effect on Cell Viability in Coculture System of BV2 Cells and PC12 Cells

To examine the effect of microglial-derived factors on neurons, BV2 cells and PC12 cells were cocultured in transwell chambers containing DMSO, Ala, LPS, and LPS + Ala groups. Furthermore, a PC12 + LPS group without BV2 cells was designed separately to exclude the direct effect of LPS on PC12 cells. Flow cytometry and immunofluorescence were used to evaluate cell apoptosis and necrosis. The early apoptotic cells were annexin V-positive and PI-negative, the late apoptotic cells were annexin V-positive and PI-positive, and total apoptotic cells were composed of early apoptotic cells and late apoptotic cells. Necrotic cells were annexin V-negative and PI-positive, and normal cells were annexin V-negative and PI-negative. The apoptotic rate is the percentage of apoptotic cells of the total cells, and necrotic rate is the percentage of necrotic cells of the total cells. Prior to this, cell viability of the PC12 cells treated with Ala was assessed using the MTT assay. As shown in Figure 5A, cell viability was not by Ala treatment at a concentration of 2 µM for 24 h. Then, in the coculture system, 24 h after the addition of LPS, we observed that a small number of cells underwent apoptosis and necrosis in the DMSO group (apoptotic rate = 3.28% ± 0.75%; necrotic rate = 0.04% ± 0.03%), the PC12 + LPS group (apoptotic rate = 3.10% ± 0.39%; necrotic rate = 0.07% ± 0.07%), and the Ala group (apoptotic rate = 3.25% ± 0.48%; necrotic rate = 0.07% ± 0.05%). LPS did not lead to the death of PC12 cells in the absence of BV2 cells (PC12 + LPS group), but a considerable proportion of cells in the LPS group underwent apoptosis and necrosis (apoptotic rate = 17.29% ± 2.80%; necrotic rate = 1.73% ± 0.57%). With the treatment of Ala, cell apoptosis and necrosis were significantly inhibited (apoptotic rate = 12.66% ± 1.87%; necrotic rate = 1.41% ± 0.39%) (Figure 5B,C).

### 3.6. Ala Alleviated Cerebral Ischemia-Reperfusion Injury in the MCAO/R Rat Model

It has been reported that the NF-κB and MAPK signaling pathways accelerate the inflammatory process in cerebral ischemia-reperfusion injury [32,33,34]. This process is consistent with the inflammatory response signaling pathway that is induced by LPS. So, to assess the effect of Ala on neuroinflammation in cerebral ischemia-reperfusion injury in vivo, we established an MCAO/R rat model. The cerebral infarct volume after ischemia-reperfusion injury was measured by TTC staining 24 h after the addition of Ala. Few infarct areas were observed in brain samples of the sham group and Ala group, whereas large infarct areas were observed in the MCAO/R group. Importantly, Ala decreased the cerebral infarct area in the MCAO/R + Ala group (Figure 6A). Data from the statistical analysis of the cerebral infarct volume provided concrete evidence that Ala administration decreased the infarct volume from approximately 50% in the MCAO/R group to approximately 25% in the MCAO/R + Ala group (Figure 6B). Furthermore, we allocated neurological deficit scores to the animals before sacrificing them. Figure 6C shows that the sham animals rarely exhibited evidence of neurological deficit, whereas the MCAO/R animals that suffered from ischemia-reperfusion injury displayed behavior that was characteristic of neuron damage and had relatively high neurological deficit scores. These high neurological deficit scores were lessened to a great extent after Ala administration. To assess brain edema in the brain region of animals with ischemia-reperfusion injury, we analyzed the BWC for each group. The BWC was statistically significantly higher in the brain samples from the MCAO/R group than in those from the MCAO/R + Ala group. The BWC in the sham group and Ala group without ischemia-reperfusion injury did not differ (Figure 6D).

### 3.7. HE Staining, TUNEL Staining, and Nissl Staining

Brain histopathology was analyzed in each group following ischemia-reperfusion injury: Specifically, the hippocampus and penumbra regions were assessed for morphological change. In the ischemic core, tissue and cells were largely lost, and connectivity cavities appeared between cells. The cells in the penumbrae were swollen. Furthermore, ambiguous or invisible nuclei, nuclear vacuolization, and cytoplasmic cavities in the MCAO/R group were present and significant. In the MCAO/R group, pyramidal neurons of the hippocampi were markedly reduced. Beyond all doubt, these effects were absent from the sham group and Ala group (Figure 7A,B). Further, pathological damage was alleviated by Ala treatment. According to Nissl staining, the number of normal neurons in the ischemic penumbrae was significantly higher in the MCAO/R + Ala group (1112 ± 168.9 per mm^2^) than in the MCAO/R group (809.4 ± 151.4 per mm^2^) (Figure 7C).

Figure 7D shows the results of TUNEL staining. The number of TUNEL-positive cells after ischemia reperfusion was dramatically higher in the sham group compared to in the Ala group. Thus, Ala prevented an increase in TUNEL-positive cells. The proportion of TUNEL-positive cells was remarkably reduced from approximately 40% in the MCAO/R group to approximately 20% in the MCAO/R + Ala group (*p* < 0.01) (Figure 7E).

### 3.8. Proinflammatory Factors in the MCAO/R Rat Model

The levels of IL-1β, IL-6, TNF-α, NO, and PGE_2_ were detected in the MCAO/R rat model (Figure 8). Statistical analysis demonstrated that the samples from the MCAO/R group had higher levels of IL-1β, IL-6, TNF-α, NO, and PGE_2_ than the sham group and Ala group samples did. The levels of these factors were substantially decreased in the MCAO/R + Ala group. Our data suggest that Ala decreased the levels of proinflammatory factors in vivo.

## 4. Discussion

Ala has been reported to exhibit significant anticancer effects by modulating the activity of the NF-κB and MAPK signaling pathways [26,27]. To our surprised, NF-κB and MAPK signaling pathways are also the major mechanisms of injurious neuroinflammation by upregulating the expression expression of proinflammatory factors to cause cell apoptosis or necrosis. Therefore, Ala might alleviate neuroinflammatory injury through its inhibitory effect on the NF-κB and MAPK signaling pathways. The inflammatory response participates in the pathophysiological mechanisms of CNS injury. Microglia occupy a pivotal position in the CNS. Sustained activated microglia contribute to the inflammatory response under the pathological conditions that cause apoptosis and necrosis in neurons that injure the CNS [35]. Therefore, in this study, microglia cultured with LPS could be used as a neuroinflammatory model for a pharmacodynamics study of Ala [36].

In this study, BV2 cells and PC12 cells were respectively used as microglia and neuron cell models for the investigation of the anti-inflammatory effect of Ala. Interactions between proinflammatory factors are involved in the occurrence, development, and outcome of neuroinflammatory injury [37]. After the cells were cultured with LPS, proinflammatory factors were expressed in the BV2 cells and caused necrosis and apoptosis of the PC12 cells. All of the results in the study indicate that this in vitro model of neuroinflammatory injury was successfully constructed.

To our astonishment, the proliferation and ameboid morphological changes in BV2 cells that were induced by LPS were significantly inhibited by Ala administration, which reduced the number of cells generating inflammatory responses. Besides this observation, we discovered that IL-1β and IL-6, as representative proinflammatory cytokines, remained inhibited 24 h after Ala administration, but Ala had no significant effect on TNF-α. These results indicate that Ala might strongly influence the regulation of IL-1β and IL-6, rather than TNF-α. As a result of microglia activation, iNOS and COX-2 were overexpressed. These proteins activate the downstream target proteins NO and PGE_2_, which strengthen the inflammation response that is associated with both acute and chronic inflammation [38]. Consistent with the results of previous studies, NO and PGE_2_ were upregulated in our experiment when induced by LPS, but were expressed at lower levels after Ala treatment. These results support our premise that Ala attenuates neuroinflammatory injury by downregulating proinflammatory factors.

It has been reported that the TLR4/NF-κB signaling pathway accelerates the inflammatory process in neuroinflammatory injury [39,40,41]. NF-κB is a nuclear transcription factor that regulates the expression of numerous genes that are responsible for regulating apoptosis, tumorigenesis, inflammation, and various autoimmune diseases [42]. Recent reviews have indicated that Toll-like receptors (TLRs) are related to NF-κB activation. As a type of pattern recognition receptor (PRR), TLRs play a critical role in transducing extracellular information into intracellular cascade reactions and triggering the inflammatory response in neuroinflammation [43]. Moreover, we found that p-IκBα and p-p65 were highly expressed and that IκBα was detected at low levels in BV2 cells after LPS treatment: Ala inhibited the degradation of IκBα but had no significant effect on p65. The p65 subunit of the activated NF-κB translocated from the cytoplasm to the nucleus, where it induced the expression of downstream target genes, including COX-2, iNOS, IL-1β, and IL-6 [44,45]. Increasingly, more research has demonstrated that the NF-κB p65 subunit is primarily retained in the cytoplasm of unstimulated BV2 cells but translocates to the nucleus when these cells are stimulated with LPS [32,46]. As shown in Figure 4C, this phenomenon, whereby the p65 subunit translocates to the nucleus, was observed in this study, and Ala inhibited this process. All of these findings suggest that Ala mediates inflammatory gene expression by inhibiting the degradation of IκBα, thus preventing the translocation of the p65 subunit to the cytoblast.

Similarly to NF-κB, AP-1 translocates to the nucleus to perform transcriptional activation when it is activated as a result of the amplified cascade reaction in the MAPK signaling pathway. The translocation of AP-1 depends on the phosphorylation of p38, ERK, and JNK [30]. Ryan et al. found that AP-1 exerts either pro- or anti-apoptotic effects, depending on the level of cell stress [47,48]. In this study, the enhanced phosphorylation of ERK and JNK promoted the translocation of a large proportion of activated AP-1 to the nucleus after BV2 cells were stimulated with LPS for 1 h. However, p38 did not change at all. To our excitement, p-JNK and p-ERK were continuously decreased, depending on the Ala pretreatment. These results show that the downregulation of phosphorylated ERK and phosphorylated JNK inhibited the activity of the MAPK signaling pathway. From the results of these in vitro experiments, we can conclude that Ala can reduce the expression of neuroinflammatory cytokines in BV2 cells stimulated by LPS.

Neuroinflammation is an important and harmful mechanism in cerebral ischemia-reperfusion injury [49,50]. It has been reported that the TLR4/NF-κB signaling pathway and the TLR4/MAPK signaling pathway has an important relationship with neuroinflammation in the MCAO/R model, which have the similar molecular mechanisms as the LPS-activated expression of proinflammatory factors in microglia. Moreover, previous studies have confirmed that neuroinflammatory lesions also activate microglia in the brain’s other sites to upregulate the expression of inflammatory chemokines [51]. Therefore, in this study, an in vivo investigation was performed in MCAO rat models. The results indicate that Ala not only reduced the infarct and water content of the brain tissue but also maintained the structure of normal brain tissue and suppressed neuronal apoptosis and necrosis. This improvement in brain injury might be attributed to the inhibitory effect of Ala on neuroinflammation. Therefore, Ala could be a promising candidate for ameliorating cerebral ischemia-reperfusion injury through its anti-neuroinflammatory effect.

## 5. Conclusions

In summary, the results of our investigation suggest that Ala inhibited the LPS-activated expression of proinflammatory factors by modulating the NF-κB and MAPK signaling pathways. Ala also blocked LPS-induced activation of COX-2 and iNOS as well as their downstream molecules NO and PGE_2_, and this was observed in both in vivo and in vitro experimental models. When PC12 cells were cocultured with LPS-activated BV2 microglia, pretreatment with Ala ameliorated the damage to PC12 by reducing apoptosis and necrosis. Ala may have exerted its neuroprotective effect by inhibiting the overactivation of microglia. All of our results support the premise that Ala may be a prospective agent for the treatment of neuroinflammatory injury.

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
