# Peer review of "Anti-Neuroinflammatory Effect of Alantolactone through the Suppression of the NF-κB and MAPK Signaling Pathways"

_cells, 2019, doi:10.3390/cells8070739_

Round 1
Reviewer 1 Report
This study by Li et al. describes anti-inflammatory effects of the naturally occurring molecule Alantolactone in neural cells. Alantolactone is a known anti-inflammatory molecule that has previously been shown to attenuate both the MAPK and NF-κB signaling pathways. The authors hypothesise that pre-treatment with Alantolactone may offer protection for inflamed neuronal cells. An in vitro co-culture system of BV2 cells and PC12 treated with lipopolysaccharide (LPS) was used to mimic neuroinflammation. The study shows that pre-treatment with Alantolactone downregulated the LPS-induced expression of proinflammatory factors and protected the cells from both apoptotic and necrotic cell death. The authors suggest that this protection is due to the suppression of the transcriptional program induced by both the NF-kB and MAPK signalling pathways. Finally, the authors used an in vivo model to demonstrate Alantolactone can reduce the damage caused by inflammation induced by ischemia/reperfusion.
General Comments
This is an interesting and well-controlled study that will be of interest to the field, however I have outlined some general comments below that I feel need to be addressed before the work is published.
1. Scheme 1 indicates that MAP kinase kinase kinase is required for Tak-mediated activation of IKK, however Tak1 is a known IKK kinase.
2. In Figure 1 b-d could the authors explicitly state the concentration of Ala used in either the text or legend.
3. The immunoblot data in Figure 4A shows a minimal (if any) effect of Ala on Ikba stability or phosphorylation in response to LPS. Densitometry quantification on the experimental repeats may indicate if there is any Ala dependent suppression of LPS-induced Ikba degradation.
4. The p-p38 blot in Figure 4B is inconclusive, could the authors provide a better blot and indicate the proposed phospho band.
5. In contrast to Figure 4A, Figure 4C does show a suppression of RelA nuclear translocation in Ala treated cells. Could the authors quantify this as a robust measure of NF-kB activation. Also for Fig 4D.
Minor
6. Line 51 – use correct name for Ikba
7. Line 52-56. P65 is generally in complex with p50 – may be worth defining
8. Figure 2. Labelling confusing – consider replacing lower case letters with numbers
9. Figure 8C – gene name is not present on graph.
Author Response
Reviewer 1.
This study by Li et al. describes anti-inflammatory effects of the naturally occurring molecule Alantolactone in neural cells. Alantolactone is a known anti-inflammatory molecule that has previously been shown to attenuate both the MAPK and NF-κB signaling pathways. The authors hypothesise that pre-treatment with Alantolactone may offer protection for inflamed neuronal cells. An in vitro co-culture system of BV2 cells and PC12 treated with lipopolysaccharide (LPS) was used to mimic neuroinflammation. The study shows that pre-treatment with Alantolactone downregulated the LPS-induced expression of proinflammatory factors and protected the cells from both apoptotic and necrotic cell death. The authors suggest that this protection is due to the suppression of the transcriptional program induced by both the NF-kB and MAPK signalling pathways. Finally, the authors used an in vivo model to demonstrate Alantolactone can reduce the damage caused by inflammation induced by ischemia/reperfusion.
This is an interesting and well-controlled study that will be of interest to the field, however I have outlined some general comments below that I feel need to be addressed before the work is published.
1. Scheme 1 indicates that MAP kinase kinase kinase is required for Tak-mediated activation of IKK, however Tak1 is a known IKK kinase.
------Thank you very much, we have corrected the scheme 1 in revised manuscript.
2. In Figure 1 b-d could the authors explicitly state the concentration of Ala used in either the text or legend.
------Thanks very much for your advice, the concentrations of Ala have been added in the revised manuscript.
3. The immunoblot data in Figure 4A shows a minimal (if any) effect of Ala on Ikba stability or phosphorylation in response to LPS. Densitometry quantification on the experimental repeats may indicate if there is any Ala dependent suppression of LPS-induced Ikba degradation.
------Thank you very much, the intensities of these protein bands in figure 4 were quantified by Image J software. And it could be observed that the Ala could significant inhibit the degradation of IκBα. The results have been added in the revised manuscript.
4. The p-p38 blot in Figure 4B is inconclusive, could the authors provide a better blot and indicate the proposed phospho band.
------Thank you very much. The p-p38 in the LPS-activated BV2 cells model could not be detected in our experiments, so that the p-p38 blot in Figure 4B is inconclusive. The results indicated that LPS and Ala had no effect on p38 phosphorylation.
5. In contrast to Figure 4A, Figure 4C does show a suppression of RelA nuclear translocation in Ala treated cells. Could the authors quantify this as a robust measure of NF-kB activation. Also for Fig 4D.
------Thank you very much. We further to make the quantitative analysis by the fluorescence intensity detection by Image J software. And we could easily observed the significant difference between each group. The results and the discussions have been added in revised manuscript.
Minor
6. Line 51 – use correct name for Ikba
------Thank you very much. We have corrected the name of IκBα in the revision.
7. Line 52-56. P65 is generally in complex with p50 – may be worth defining
------Thanks very much for your suggestion. p65/p50 heterodimer is the most common in NF-κB family. In this study, the p65 was the subunit of NF-κB family which could regulate the binding of p50 with DNA to activate the downstream reaction. So, the p65 was used as the marker to investigate the activity of NF-κB signaling pathway. The p65 has been defined in revised manuscript.
8. Figure 2. Labelling confusing – consider replacing lower case letters with numbers
------Thanks very much for your advice, the clear labels have been modified in revised manuscript.
9. Figure 8C – gene name is not present on graph.
------Thanks very much. The gene name has been added in figure 8C in revised manuscript.
Reviewer 2 Report
Comments are listed in the attached document.

Author Response
Reviewer 2.
Thank you very much for taking your time to review our manuscript and giving us the valuable comments.
1. Language problems.
------Thank you very much. We have carefully corrected the grammatical and structural errors in revised manuscript, and the language has been further polished. The manuscript has been kindly undergone English language editing by MDPI.
2. Line 19, what does this mean?
------Thank you very much. The co-culture system of BV2 cells and PC12 cells in this study were used as the neuroinflammatory model for evaluating the effect of Ala. We have modified the description in the revised manuscript.
3. Line 27, rephrase this statement.
-------Thanks for your advice, we have rephrase the statement in the revised manuscript.
4. Line 37, especially from natural products. some relevant citations could be included in building up the introduction. And rephrase.
https://www.frontiersin.org/articles/10.3389/fphar.2017.00397/full
https://link.springer.com/article/10.1007/s11418-011-0622-y
------- Thank you very much. We have rephrased the description in the revised manuscript. And the previous superior works have been cited to enrich our background.
5. Line 43-44, rephrase.
------- Thank you very much. The description has been rephrased in revised manuscript.
6. Some grammar mistake in line 47, line 50, line 58 and line 60.
------- Thank you very much. We feel so sorry for these stupid mistakes. We have corrected the all the mistakes in revision carefully.
7. Line 68, proven to be able in exerting its inhibitory effect on growth of tumor.
------Thank you very much. We have corrected the description in the revision.
8. Line 72, rephrase.
------Thank you very much. We have corrected the description in the revision.
9. Line 74, not a novel.
------Thank you very much. Ala is a natural small-molecule agent, but not a novel one.
10. Line 75 and line 76, rephrase.
------Thank you very much. The description has been modified as this: in this study, we explored the anti-neuroinflammatory properties of a natural small-molecular agent (Ala) in vitro and in vivo, and further to made a detail investigation for its pharmacodynamic effect and mechanism.
11. Line 79-line 84, some mistakes should be corrected.
------Thank you very much. We have corrected the mistakes in revised manuscript.
12. Line 87, wrong words.
------Thank you very much. We have corrected the “classic” to “classical”.
13. Line 118-119, Ethics proof.
-------Thank you very much. The study was the part of project of the National Natural Science Foundation of China, the ethic approval file has been provided in the author statement.
14. Line 137-138, realign.
-------Thank you very much. The structure of this manuscript have been modified in revision.
15. Line 158. Event count of each assays should be named somewhere in methodology.
-------Thank you very much. The Event count of each assays has been added in the methods.
16. Line 217. Analysis.
-------Thank you very much. “flow cytometry” has been corrected as “flow cytometry analysis”.
17. Figure 1, The concentration of Ala used in this part of assays? Ala treatment alone whether is exhibiting a significant different changes between DMSO and LPS alone?
-------Thank you very much. The concentration of Ala has been described in legend. The difference analysis of between DMSO and LPS alone were added in the figure 1C, the results showed that there was a significant different change between DMSO and LPS alone.
18. Figere1D, what statistical analysis was being done? The error bars are seem to be overlapping but still is showing a significant in levels of changes?
------- Thank you very much. We the statistical analysis performed by prism version 6.0. The P value between the LPS group and LPS+Ala group was 0.007 less than 0.01, which indicated the significant difference.
19. Line 363-364.
------- Thank you very much. The Standard deviation has been added in revised manuscript.
20. Figure 5, Please follow the way of naming the quadrant; early apoptosis and late apoptosis especially.
https://www.sciencedirect.com/science/article/pii/S0378874115002317.
------- Thank you very much. The ways of naming the quadrant have corrected follow the previous superior work that has been cited in revised manuscript. And early apoptosis and late apoptosis have been specifically marked.
21. Line439-440, rephrase.
------- Thank you very much. We have made the modification in revised manuscript.
22. Line443-447, rephrase.
------- Thank you very much. The description has been further polished in revised manuscript.
23. Line448-449, rephrase.
------- Thank you very much. We have rephrased the sentence.
24. Line 497-450, rephrase.
------- Thank you very much. The description has been further polished in revised manuscript.
25. Line 505, rephrase.
------ Thank you very much. The description has been corrected as “Therefore, Ala would be a promising candidate for ameliorating the cerebral ischemia-reperfusion injury by its anti-neuroinflammatory effect.”
26. Conclusion section. Correct some descriptions.
------ Thanks for your suggestion. We have modified the descriptions follow your suggestion.
27. Replicate of in each experiment.
------- Thank you very much. The replicate of each experiment has been added in revised manuscript.
Finally, thank you for all the important comments and suggestions from you and the reviewers, which are valuable in improving the quality of our manuscript. Thank you again for your contributions and consideration.
Best regards!
Rui Tan